# A Thermopile Device with Sub-Wavelength Hole Arrays by CMOS-MEMS Technology

**DOI:** 10.3390/s21010180

**Published:** 2020-12-29

**Authors:** Chi-Feng Chen, Chih-Hsiung Shen, Yun-Ying Yeh

**Affiliations:** 1Department of Mechanical Engineering, National Central University, Taoyuan City 32001, Taiwan; ccf@cc.ncu.edu.tw (C.-F.C.); j9038592@ms38.hinet.net (Y.-Y.Y.); 2Department of Mechatronics Engineering, National Changhua University of Education, Changhua City 50007, Taiwan

**Keywords:** sub-wavelength, sub-wavelength hole arrays, thermopile, CMOS-MEMS, infrared radiation, infrared sensors, infrared absorbance, infrared absorption efficiency

## Abstract

A thermopile device with sub-wavelength hole array (SHA) is numerically and experimentally investigated. The infrared absorbance (IRA) effect of SHAs in active area of the thermopile device is clearly analyzed by the finite-difference time-domain (FDTD) method. The prototypes are manufactured by the 0.35 μm 2P4M complementary metal-oxide-semiconductor micro-electro-mechanical-systems (CMOS-MEMS) process in Taiwan semiconductor manufacturing company (TSMC). The measurement results of those prototypes are similar to their simulation results. Based on the simulation technology, more sub-wavelength hole structural effects for IRA of such thermopile device are discussed. It is found from simulation results that the results of SHAs arranged in a hexagonal shape are significantly better than the results of SHAs arranged in a square and the infrared absorption efficiencies (IAEs) of specific asymmetric rectangle and elliptical hole structure arrays are higher than the relatively symmetric square and circular hole structure arrays. The overall best results are respectively up to 3.532 and 3.573 times higher than that without sub-wavelength structure at the target temperature of 60 °C when the minimum structure line width limit of the process is ignored. Obviously, the IRA can be enhanced when the SHAs are considered in active area of the thermopile device and the structural optimization of the SHAs is absolutely necessary.

## 1. Introduction

The temperature sensor is a kind of sensor that was developed very early, widely used and most commonly used. Especially in this century, there are many diseases that endanger human life, such as influenza, severe acute respiratory syndrome coronavirus (SARS-CoV), and coronavirus disease 2019 (COVID-19) [1]. Fever is one of the main symptoms of these diseases. Without avoiding virus infection, non-contact temperature sensors are gradually becoming [2]. In addition, the non-contact temperature sensing device or non-contact temperature measuring instrument also has the application characteristics of measuring the surface temperature of moving objects, small measurement targets, small heat capacity or rapid temperature changes, and measuring the temperature distribution of the temperature field [3,4]. According to Stefan-Boltzmann law, infrared (IR) sensors (IR sensors for short) are often used as non-contact temperature sensors [3]. Mainly served as the temperature measurement device, the thermopile delivers the output voltage in response to remote temperature, such as an infrared thermometer widely used by medical professionals to measure body temperature, or in a thermal accelerometer to measure the temperature curve in the sealed cavity of the sensor. In particular, it is also very suitable for remote temperature sensing [5] and non-dispersive infrared sensing (NDIR) gas detection [6,7].

These IR sensors mainly include thermal-type detectors such as thermopile, bolometer or pyroelectric detector to detect the change of temperature on the element for measurement through thermal radiation exchange between the targets and environment. The temperature under thermal equilibrium will then be converted into a measurable detector electrical signal. In particular, the sensing wavelength range of these thermal radiation generally covers a very wide infrared spectral range.

In recent decades, many research and development trends have been realized in standard CMOS (complementary metal oxide semiconductor) processes for the fabrication of IR sensors. Due to the maturity of the process and compatibility with standard CMOS technology, the literature reviews the use of CMOS-MEMS technology to promote the development of IR components, which has been proven to be particularly effective in the development of SOC (system on chip) technology [8,9,10]. Therefore, this research aims to improve the sensitivity of CMOS-MEMS IR sensors to realize the development of high heat radiation absorption technology [11,12,13,14,15]. CMOS-MEMS thermopile is composed of multiple thermocouples (in series), and the weak voltage signal generated by each pair of thermocouples will be added and output to the next stage of amplifying sensing circuit. The final output signal can be used for measuring the amount of IR radiation absorbed by the detector [16,17].

Among these studies, the research issues of infrared absorption technology mainly are focused on improving the absorption efficiency and sensitivity of the sensing film. Among these research, the interference-type ultra-thin metal film or quarter-wavelength structure [18,19] above the sensing area are fabricated through the sputtering or evaporation process, which can enhance the absorption of IR heat radiation within a specific range. This thin film process is compatible with semiconductor processes and proceeded after the fabrication of device. Related similar processes also include the use of thermal coating of deposition to produce high-porosity or gold-black structures to improve the bandwidth of IR absorption coatings [20]. The advantage of this technology is a broad absorption spectrum, so it provides a better sensing signal for commonly used for the thermal radiation thermometers. At the same time, it is reported that some studies have also focused on the deposition and filling of the polymer coating of particles [21]. In addition, the exploration and research of new materials with high spectral absorption are still proceeded.

In the above studies, although the way of realization and production is different, the enhanced absorption layer improves the sensing efficiency of thermal radiation. Most of these studies have the following characteristics: (i) higher responsively and wider absorption spectrum, (ii) The structural design has a small heat capacity, (iii) the heat dissipation structure has a better conductivity, (iv) the device has long-term stability and repeatability in operation, (v) the manufacturing process of the device is compatible to the material deposition [22]. These characteristics show the main issues to be considered in the development of infrared thermal radiation sensing films.

In these research topics, due to the continuous development and research of new materials and the innovative progress of research, new features and methods are provided for the development of sensing materials in infrared components. In 2013, the research work by Mikyung Lim et al. proposed to coat a single atomic layer of graphene on a doped silicon substrate. This study revealed that it can increase the radiant heat flux between two plates [23] and it also shows that the new material, graphene, has a sp^2^-hybrid honeycomb two-dimensional carbon lattice composed of conjugated hexagonal cells, and shows a higher heat radiation absorption capacity, which is a higher emissivity performance.

Generally, when applied to a remote temperature measurement ranging from room temperature to hundreds of degrees, the main sensing radiation wavelength of the IR sensor covers a range of approximately several microns to several tens of microns. In other words, for CMOS-MEMS technology, the structure pattern and size corresponding to the wavelength response of the IR sensor is estimated in the sub-wavelength range.

The sub-wavelength structure, which is usually arranged in repeating patterns on the surface or in the material, will affect the phenomenon of the propagation light waves which is firstly proposed, fabricated and verified by C. H. Shen [24]. More research may even cause the appearance of some special optical phenomena [25,26,27,28,29,30], such as negative refraction [31,32], superprism [33], anomalous reflection [34]. Such materials with special structures are usually called photonic crystal [25,26], optical magnetism [27], or optical metasurface [28]. They can achieve higher characteristics than those achieved by natural materials, and even achieve characteristics that are not found in natural materials [34,35]. The desired characteristics must usually be realized by numerical methods to design the shape, size, direction and array arrangement of the structure patterns. Common methods include finite-difference time-domain (FDTD) method [36,37,38,39], plane wave expansion method [40], multiple scattering method [41], and transfer-matrix method [42].

Some patterns of sub-wavelength hole arrays (SHAs) located in the active area are proposed to generate special phenomenon for the propagation light waves in a designated wavelength range. The effect of extraordinary optical transmission (EOT) at specific wavelengths is discovered in metal films with two-dimensional sub-wavelength cylindrical-hole arrays [43]. It is found that the strength of the effect depends on the geometric factors [44,45,46], such as the periodicity of the hole array, the thickness of the film, and the shape and arrangement of the holes. The free-standing perforated metal films are fabricated to explore this EOT effect. The transmission enhancement of terahertz (THz) radiation through the appropriate SHAs in highly doped silicon wafer [47] or the n-type silicon [48] or through the thin metallic films patterned with sub-wavelength hole arrays on silicon wafers [49] is presented and demonstrated. The extraordinary transmission of THz the far-field and near-field physics of extraordinary THz transmitting through SHAs under different illumination and detection conditions are investigated [50]. The experimental results are consistent with that of the numerical results, that is, this computationally efficient modeling tool can be used to predict the response of extraordinary transmission structures in real situations. In addition, the investigation of fluorescence enhancement is realized by the periodic array of sub-wavelength apertures in a metal film coated in fluorescing molecular monolayer [51]. The measurement results show that the fluorescence enhancement is about 30–40 times and the total fluorescence yield is increased by about 15–20 times.

Based on CMOS-MEMS technology, an integrated and miniaturized thermopile sensor with various SHAs is proposed [24], and the influence of geometry on the infrared absorption efficiency (IAE) of some different types of SHAs is further studied [24,52]. The square and hexagonal patterns are considered to arrange subwavelength rectangular hole array (SRHA) [52]. The FDTD simulation results shows the cases of square arrangement (SA) are better than those of hexagonal arrangement (HA) in similar conditions. The proposed thermopiles with SA-SRHA are manufactured and measured. The measurement and simulation results are obvious similarity. In addition, we find an interesting phenomenon that several special subwavelength columnar structures are added in the best rectangular-hole structure obtained in [24] and the optimal simulation results can increase by more than 14% [53].

In this article, we will systematically discuss the research results of these thermopile sensors, and study the structural optimization of several SHAs through numerical methods. The investigated SHA patterns are three types of hole shapes including rectangular, square, and circular, referred to as SRHA, SSHA, and SCHA, and two kinds of hole arrays with SA and HA. The top-view sketches of (a) three investigated structure patterns and (b) the hole arrangements described by use of circular holes as an example are shown in Figure 1. In the simulation model, we set the plane shown in Figure 1 to the *X*-*Y* plane and set the *X*-axis to the horizontal direction. Among these three structures, except for the rectangle, it is seen from Figure 1a that their hole widths are the same in the *X*-axis and *Y*-axis directions. So that, for the types of SSHA and SCHA, the hole widths are represented by *w_S_* and *d*, respectively. For SRHA type, the hole widths in the *X*-axis and *Y*-axis directions are respectively represented by *w_Rx_* and *w_Ry_*. Here the hole wall widths defined as the minimum width of the wall between two adjacent holes are assumed to be the same in any direction of the *X*-*Y* plane and are represented by *ρ_R_*, *ρ_S_*, and *ρ_C_,* respectively, for three types of SRHA, SSHA, and SCHA.

## 2. Research Preliminary

### 2.1. Configuration and Theory of CMOS Compatible Thermopile

In order to further explore the infrared absorbance (IRA) characteristics of SHA, a suspended thin film structure containing a thermopile is designed and fabricated. After the standard CMOS processes, the silicon substrate under the floating structure will be further isotropic etched and removed, so that the thin film structure above the silicon cavity will be suspended. The thin film of the thermopile has a smaller heat capacity and solid thermal conductance. The thorough heat exchange mechanism is shown in Figure 2 [24,52]. In general, the dynamic thermal behavior of the infrared sensing element is described by the heat equation which includes the heat radiation exchange mechanism between the infrared radiation source and the sensor and also solid conduction, convection which are all described in Equation (1). For different wavelengths, the spectral absorption of thermal radiation for SHA is noted as *R*(*λ*). The absorption of the rest area of cantilever beam structure is written as *R_m_*. The corresponding temperatures for hot junction, ambient environment and radiation source are denoted as $T_{h}$, $T_{a}$ and $T_{b}$, separately. The light path between the radiation source and sensor is quite complicated and the geometrical factor of the path is used as a constant $S_{o}$. $\varepsilon_{b}$ and $\varepsilon_{a}$ are the emissivity of radiation source and the sensor. H is the heat capacitance of thermopile membrane and the h is the convective heat transfer coefficient. $A_{a}$, $A_{m}$ are the active area and the rest area of cantilever beam structure and $A_{o}$ is the total area of cantilever beam structure.
(1)$$\begin{array}{l} H\frac{dT_{h}}{dt}+G_{s}(T_{h}-T_{a})+hA_{o}(T_{h}-T_{a})=P_{e}+\int_{0}^{\infty }S_{o}\varepsilon_{b}\sigma A_{a}\frac{2hc}{\lambda^{3}}\frac{R(\lambda )}{e^{hc/\lambda kT_{b}}-1}d\lambda \\ +\int_{0}^{\infty }S_{o}\varepsilon_{b}\sigma A_{m}\frac{2hc}{\lambda^{3}}\frac{R_{m}}{e^{hc/\lambda kT_{b}}-1}d\lambda -\varepsilon_{a}\sigma A_{o}T_{a}^{4} \end{array}$$

In addition to the implementation of simulation analysis, it is more practical to modify Equation (1) as a discrete expression of wavelength in Equation (2), not an integral form.
(2)$$\begin{array}{l} H\frac{dT_{h}}{dt}+G_{s}(T_{h}-T_{a})+hA_{o}(T_{h}-T_{a})=P_{e}+\sum_{i}S_{o}\varepsilon_{b}\sigma A_{a}\frac{2hc}{\lambda_{i}^{3}}\frac{R(\lambda_{i})}{e^{hc/\lambda_{i}kT_{b}}-1}\Delta \lambda \\ +\sum_{i}S_{o}\varepsilon_{b}\sigma A_{m}\frac{2hc}{\lambda_{i}^{3}}\frac{R_{m}(\lambda_{i})}{e^{hc/\lambda_{i}kT_{b}}-1}\Delta \lambda -\varepsilon_{a}\sigma A_{o}T_{a}^{4} \end{array}$$

Without external heat source $P_{e}$, Equation (2) can be further simplified and manipulated as Equation (3) and the infrared radiation exchange plays the major role of heat source in heat equation. ε is the emissivity of absorption area and especially it plays the major role of the IRA for our proposed SHAs. $G_{o}$ is the total thermal conductance including the solid and heat convection which conduct the heat to the environment at the ambient temperature $\;T_{a}$ i.e., $G_{o}$ = $G_{s}$ + $G_{c}$.
(3)$$H\frac{dT_{h}}{dt}+G_{o}(T_{h}-T_{a})=\;\varepsilon \sigma A(T_{b}^{4}-T_{a}^{4})$$
(4)$$\tau =\frac{H}{G_{o}}=\frac{1}{\omega_{c}}=\frac{1}{2\pi f_{c}}$$

The signal is inversely proportional to the total thermal conductance including the solid and heat convection which conduct the heat to the environment at the ambient temperature. A better thermal isolation will reduce the solid conduction which give the contribution to the signal depending on the ratio of the solid conduction and heat convection. Beside of the spectral response, the response time and frequency response bandwidth $f_{c}$ of the thermal sensor are functions of *H* and *G*, which can be estimated and measured. The frequency response curve of infrared radiation is modulated by a mechanical chopper first, and then the thermal time constant can be measured and derived, expressed as the following Equation (4).

### 2.2. Process Description

Based on the 0.35 μm 2P4M CMOS-MEMS process in TSMC [16], the thermopiles with specific SHA are designed and fabricated [24,52]. To accurately construct the SHA design model, it is necessary to understand the 0.35 μm 2P4M CMOS-MEMS process. Therefore, we first describe the 0.35 μm 2P4M CMOS-MEMS process.

Using TSMC 0.35 µm 2P4M CMOS-MEMS, the CMOS thermopile is designed and fabricated as the testing structure which is shown in Figure 3. According to the standard CMOS process in TSMC (TSMC), polysilicon and aluminum in the standard CMOS process are usually used as the structural material of the thermocouple that constitutes the thermopile, as shown in step 1. As shown in Figure 2, this design uses the post-CMOS process to form the etching window of the silicon substrate and the subwavelength structure SHA which is expected to absorb infrared heat radiation. MEMS process design uses two process steps, RLS and RLSSI, to perform the subsequent removal process of etching the SiO_2_ layer and the silicon substrate. The SHA pattern is formed by the PAD process to firstly remove the Si_3_N_4_/SiO_2_ layers and the subsequent RIE etching for the SiO_2_ layer on the active area in the RLS process. By using the RLSSI process, vertical and lateral RIE etching of the silicon substrate is performed under the film. Therefore, the sensing area is floated and filled with arrays of etching hole as the periodic refractive index waveguide. After removing the silicon substrate under the cantilever beam, a thin floating structure layer with a thickness of 7 μm is finally left as the suspension structure. The thermocouple is designed with a structural made of n-type polysilicon material, with a width of 20 μm and a length of 200 μm. The aluminum metal material with low thermoelectric coefficient has a width of 0.5 μm and is deposited on top of n-type polysilicon. Finally, the thin cantilever beam is suspended on the etched cavity with low thermal conductivity, which is used as a test platform for the analysis of SHA. Moreover, to investigate the absorption performance of SHA on CMOS compatible thermopile, a thermopile without SHA was also fabricated and simulated.

## 3. Simulation Tool Building and Preliminary Experimental Verification

### 3.1. Simulation Tool Building

To search the SHA geometric parameters with better results, an accurate and efficient simulation tool is necessary. Based on the FDTD method, the simplified simulation model is setup by following the design rules of the TSMC 2P4M CMOS-MEMS process and considering the computational efficiency [24,52]. The so-called simplified simulation model that the simulation model only considers the IR light wave propagation on the active area and the materials of active area are simplified to consider only one layer of SiO_2_. In addition, because SiO_2_ has exactly obvious absorption characteristics for IR wavelengths between 8 μm and 10 μm, it just meets the target temperature set in this study. Therefore, when the simulation IRA is proceeded, we just consider the spectral absorption of thermal radiation of SiO_2_ in the range of 8 μm to 10 μm. Figure 4 shows the sketch of the IR light wave propagation from the incident medium along *Z*-axis direction into the active area of a thermopile with SHA. The IR light wave through the active area will be gradually absorbed. If there is IR light wave that is not completely absorbed by the material, it will be transmitted to the transmission medium. Then, the sketch of the light wave propagation through the CMOS compatible thermopile with the SHAs is shown in Figure 2. The incident IR light wave propagates along *Z*-axis direction from incident medium into the active area. Some light waves reflected at the incident interface will return to the incident medium and is absorbed by the reflection detector. The remaining light waves will leave the effective area and enter the transmission medium and be absorbed by the transmission detector. Most of the IR light waves entering the incident interface will be absorbed by the medium of active area. The transmission detector is arranged in the transmission medium to absorb the remaining light waves into the transmission medium through the active area medium. By the principle of conservation of energy, the total radiant energy will be completely divided into a reflection part, an absorption part by the material, and a transmission part. Statistics and comparison of these values can help to confirm the reliability of the simulation model. This is the main reason why the reflection detector and the transmission detector are set in this model. The refractive indexes of incident medium, active-area medium, and transmission medium are denoted by *n*_0_, *n_a_*, and *n_t_*, respectively. In addition, there is a reflection problem at the finite analysis window when simulation technology is used to solve electromagnetic wave problems. In the FDTD algorithm, an artificial boundary condition called perfectly matched layer (PML) is originally proposed by Berenger et al. [54]. The PML can effectively suppress the reflection at the analysis window, so the error caused by the boundary of the simulation area can be reduced [55,56]. For the simplified simulation model, the medium of active area is only SiO_2_, and the refractive indexes *n*_0_, *n_a_*, and *n_t_*, are taken as 1, 1.42, and 1, respectively. Following the principle of conservation of energy, the entire incident radiation energy is divided into the outside of the medium, including reflection and transmission, and the part absorbed by the medium. It is seen from Figure 4 that the reflection detector and the transmission detector are used to receive the reflection and the transmission parts of radiant energy, respectively. For the total absorption of active area medium, the two statistical results are almost the same, one is directly counted in simulation tool, the other is calculated from the principle of conservation of energy.

### 3.2. Design for Preliminary Experiment

According to the design rules of the above process, the minimum structure line width is limited to 3.0 μm. Therefore, the hole width and minimum wall width are assumed as 3.0 μm. To verify the effect of SHAs, preliminary experiment is prepared and the geometric parameters are designed by the above simulation tool. The IAE is defined the ratio of IRA at target temperature is relative to that at ambient temperature. In the simulation process, the IAE is calculated for the difference between the IRA obtained at target temperature to the one obtained at ambient temperature. Here we try to control the measured room temperature at 30 °C, so that, the IAE is equal to 0 when the target temperature is 30 °C. Two hole shapes of rectangle and circle are considered and two hole arrays are square arrangements [24]. Moreover, it is assumed that the hole width and the smaller hole width are the same, that is, *ρ_R_ = w_Ry_* for SRHA and *ρ_C_ = d* for SCHA. Range and interval of geometric parameters selected during simulation are shown in Table 1. For the SRHA type, due to the wide search range of *w_Rx_*, the search is divided into two stages. First, in the preliminary search stage, the interval is taken as 3 μm. Then, a detailed search is performed near the good results and the interval is taken as 1 μm. Figure 5 show the variances of the IRA by the function of minimum wall width (*ρ_R_* or *ρ_C_*) for the thermopiles with (a) SRHA and (b) SCHA, and the thermopile without any SHA. It is obtained that the geometric parameters for the best cases of SRHA and SCHA types are *ρ_R_* = 3.5 μm, *w_Rx_* = 15.0 μm, and *w_Ry_* = 3.5 μm, and *ρ_C_ = d* = 3.0 μm, and those IRAs are 88.73% and 76.81%, respectively.

### 3.3. Preliminary Experiment and Measurement of Trial Samples

In order to verify the impact of SHAs on thermopile equipment through simulation, several samples were considered, including four suggested thermopiles with SRHA and SCHA and thermopiles without any SHA [24]. The geometric parameters are listed in Table 2. Figure 6a,b show the SEM of sample 1 and sample 4. It can be seen that the SHA structure in the active area of the thermopile device has been successfully fabricated through the CMOS-MEMS process. In this process, an active area full of SHA is fabricated, and a cantilever beam structure is well established. Table 3 shows the measurement results of SHA with different geometric parameters and the corresponding simulation results of five samples. The reasons for these deviations may be due to several factors including overexposure, overetching during the processes or aberration of SEM for deformed cantilever beam.

Therefore, the characteristics of the test thermopile have been studied, and the experimental measurement setup is shown in Figure 7. The responsivity measurement uses the standard blackbody radiation source Hotech 370. This equipment provides the broad band radiation of wavelength range with a close to the ideal blackbody radiation spectrum. The experiment is set for the different target temperatures based on Hotech 370. At the same time, a calibrated infrared thermometer ST-632 is used as the comparison and confirmation of the radiation temperature. Under various target temperature conditions, the measurement results of SRHA thermopile output voltage are shown in Figure 7a. The measurement setup includes a standard infrared radiation source (IRS) and a modulated mechanical chopper system. The output signal of the thermopile is amplified by the low noise, low temperature drift chopper amplifier AD8551, and then it is transmitted to the data acquisition device NI USB-6009. The temperature of the standard IRS is set to 30–60 °C with an interval of 15 °C. In addition, in order to avoid the interference of ambient light or other background signals, a low-frequency chopper is installed before the infrared thermopile and the 5–14 μm infrared filter. To study the bandwidth of the thermopile proposed, the measurement result of the frequency response was established, as shown in Figure 7b. The frequency range is 1~150 Hz, and the bandwidth used to study the frequency response is much higher than the bandwidth of the recommended thermopile.

The measured output voltages and the simulation results normalized by the individual maximum of them at target temperature 60 °C as the function of the target temperature for the thermopiles with SRHA and SCHA and without any SHA are shown in Figure 8. One can see that the best case in Figure 8a is the SRHA of *ρ_R_* = 3.5 μm, *w_Rx_* = 15.0 μm, and *w_Ry_* = 3.5 μm, and it is the same as simulation result. Comparing Figure 8a,b, it is found that the agreement between the experimental and simulation results is good. That is, for the IAE response of CMOS compatible thermocouples with SRHA and SCHA or other similar SHA, it can be effectively predicted by using the simplified simulation model. In additional, the results of the infrared modulation measurement for thermal time constants of sensor under various conditions are obtained. Average of thermal time constant is about 5.0 ms and average cut-off frequency is about 31.9 Hz which is proved to be practical for the applications of infrared thermometer and thermal imager.

## 4. Structural Optimization of Several SHAs for IAE

The simplified simulation tool based on FDTD method is verified by the previous section and used to further study the structural optimization of several SHAs in active area of the CMOS compatible thermopile to enhance their IRA. It is found from Figure 5a that, for the rectangular structure, there are two better situations, one is that the shape is asymmetric, the other is approximately symmetric and, for the circular structure, the symmetrical shape results better within the range of our selected parameters. Therefore, three types of holes including rectangular, square, and circular, referred to as SRHA, SSHA, and SCHA, are considered for more detailed discussion and look for their individual better geometry. In addition, in the arrangement of those hole array, SA and HA are considered.

### 4.1. No Minimum Structure Line Width Limit

First, for the SRHA type, the IRAs of this thermopile are tested by adjusting *ρ_R_* at the RH widths of 15.5 μm and 5.5 μm and the results are shown in Figure 9. It is found that when *ρ_R_* is approximately equal to 1.15 μm, the IRAs of SA and HA types are the best and about 90.39% and 96.26%, respectively. Then, Figure 10 show the variances of the IRA with different *ρ_R_* by the function of *w_Ry_* for the thermopiles with SA type (a) and HA type (b) of SRHA of *w_Rx_* = 15.5 μm. One can see that the best results of SA and HA types are 95.57% and 96.55% when *w_Ry_* = 4.2 μm and 5.2 μm, respectively. Next, we continue to fine-tune the geometric conditions for better results and the results show in Figure 11. It is obtained that, for SA type, the best IRA is about 95.96% at *ρ_R_* = 1.15 μm, *w_Rx_* = 15.2 μm, and *w_Ry_* = 4.2 μm, and, for HA type, the best IRA is about 97.03% at *ρ_R_* = 1.15 μm, *w_Rx_* = 15.3 μm, and *w_Ry_* = 5.2 μm.

Then, the suitable geometric conditions of the SSHA and SCHA types are studied and the simulation results are shown in Figure 12 and Figure 13. Figure 12 show the variances of the IRA with different *ρ_S_* by the function of *w_S_* for the thermopiles with SA type (a) and HA type (b) of SSHA when *ρ_S_* = 1.10 μm, 1.15 μm, and 1.20 μm. It is found that the best IRAs of SA and HA types are about 95.53% and 96.62% at *ρ_S_* = 1.15 μm and *w_S_* = 4.3 μm, and *ρ_S_* = 1.15 μm and *w_S_* = 5.3 μm, respectively. For SCHA type, at (a) *ρ_C_* = 1.20 μm, 1.25 μm, and 1.30 μm (b) *ρ_C_* = 1.00 μm, 1.05 μm, and 1.10 μm, the variances of the IRA with different *ρ_C_* by the function of *d* for the thermopiles with SA type (a) and HA type (b) are shown in Figure 13. One can see that, for SA and HA types, the best results are about 95.97% and 97.27% at *ρ_C_* = 1.25 μm and *d* = 4.2 μm, *ρ_C_* = 1.05 μm and *d* = 5.3 μm, respectively. Then, the geometric parameters and the simulation results of the IRA and IAE normalized by the maximum of them at target temperature 60 °C for several thermopiles for the thermopiles with various optimal SHA are listed in Table 4. According to the above simulation results, an interesting result is presented, which is, for the optimal geometric patterns of the rectangular-hole and square-hole arrays, the minimum wall widths of SA and HA types are the same and their hole widths are different, and for the optimal geometric patterns of the circular-hole array, the SA and HA types are not the same. Obvious, the optimizations of geometric parameters for three investigated structure patterns can greatly improve IAE and the HA type results are significantly better than the SA type results. And, whether it is SA type or HA type, the best results of thermopiles with various SHA are only little difference, especially the types of SRHA and SCHA.

### 4.2. Minimum Structure Line width 2.5 μm

From Figure 9, it is found that the result of *ρ_R_* = 2.5 μm is better than *ρ_R_* = 3.0 μm. Moreover, according to trial production experience seen from Table 3, the actual limit of the process line width is 2.5 μm [52]. So that, for the further optimization of trial production, the minimum structure line width is taken as 2.5 μm, that is, *ρ_R_* = *ρ_S_* = *ρ_C_* = 2.5 μm. Under this condition, for each of three geometric types of thermopile, the best IAE result will be searched.

The SRHA type with a minimum wall thickness of 2.5 μm is studied in [52]. The optimal geometric parameters are obtained by the simulation tool and the variances of the IRA with different *w_Ry_* by the function of *w_Rx_* for the thermopiles with SA type and HA type of SRHA of *ρ_R_* = 2.5 μm are shown in Figure 14a,b, respectively. For SA type, the best IRA is about 89.52% at *w_Rx_* = 15.5 μm, and *w_Ry_* = 5.5 μm, and, for HA type, the best IRA is about 94.39% at *w_Rx_* = 15.5 μm, and *w_Ry_* = 5.5 μm. Next, for the types of SSHA and SCHA at minimum wall width of 2.5 μm, the variances of the IRA by the function of minimum hole width, *w_S_* or *d*, for the thermopiles with SA type and HA type are shown in Figure 15a,b, respectively. It is found that, for SSHA type, the best IRAs of SA and HA types are about 85.89% and 93.12% at *w_S_* = 5.3 μm and *w_S_* = 5.6 μm, and for SCHA type, the best IRAs of SA and HA types are about 86.75% and 93.93% at *d* = 4.6 μm and *d* = 5.1 μm, respectively. Table 5 shows the geometric parameters and the simulation results of the IRA and IAE normalized by the maximum of them at target temperature 60 °C for the thermopiles with various optimal SHA. Comparing the results for three types at minimum wall width of 2.5 μm, it is obtained that the best results of thermopiles with SRHA are significantly better than others, whether it is SA type or HA type.

### 4.3. Experimental Verification for the Better Designed Structure

To the better designed results of the above section, three trial-produced samples with various HA-SRHA are considered [52], where the X-direction hole widths of sample 1, sample 2, and sample 3 are respectively 12.5 μm, 15.5 μm, and 18.5 μm, and their hole widths in Y-direction are the same as 5.5 μm.

After process, three trial-produced samples are obtained. The sample 2 is taken as an example to verify the process results of the hole array by SEM measurement equipment and the image is shown in Figure 16. It is obtained that the structures of HA-SRHA are well fabricated for thermopile devices and both are successfully fabricated by CMOS-MEMS process. Through measurement and calculation, the geometric parameters of three proposed thermopiles are obtained and listed in Table 6. Comparing the measurement results with the original parameters, it can be found that the holes of HA-SRHA are well defined as our design. Although there are still some deviations, overall, the manufacture of those prototypes is successful. Since the geometric deviation will slightly affect the performance of the thermopile, the revised simulation based on those measurement geometric parameters is re-executed and the IAEs normalized by the maximum of them at target temperature 60 °C. Figure 17 show the measured output voltages and the simulated IAEs normalized by the individual maximum of them at target temperature 60 °C as the function of the target temperature for three trial thermopiles. Comparing Figure 17a,b, it is found that the normalized experiment results are similar to the normalized simulation results and is verified again that the simulation tool is reliable. Moreover, comparing with the thermopile without any SHA, at the target temperature of 60 °C, the IAE of the best case of HA-SRHA is up to 3.33 times higher than that without any SHA.

## 5. Other Special Structures

In this section, the minimum structure line width limit of the process is ignored. To explore the effect of the CMOS compatible thermopile with various extra subwavelength columnar structures (ESCS) in rectangular holes of the [24] best case, six designated ESCSs and look for the better geometry of the ESCSs by using the FDTD method are considered [53]. It is shown in [53] that the subwavelength rectangular-hole arrays with rectangular-columnar or elliptical-columnar structures in the hole array can be enhanced the absorption efficiency of this thermopile. Based on the results of [53], four better geometry of the ESCSs are considered in this study and the top-view sketch of four ESCSs including one rectangular column (RC), three RCs, one elliptical column (EC), and three ECs are shown in Figure 18. Here the geometric dimensions in the *X*-axis and *Y*-axis directions for the rectangular column are $\;W_{x}$ and $W_{y}$, and the ones for the elliptical column are $D_{x}$ and $D_{y}$, respectively. And, based on the requirements of structure and heat conduction, some connection structures are added to connect those ESCSs to the main structure and their values are uniformly set to 0.8 μm. The structures can be fabricated by the etching of layers and substrates beneath the floating structures.

Similarly, we hope to enhance the IRA effect of the best case in [52] by this technique, where rectangular holes of the hole width in *X*-axis direction *w_Rx_* = 15.5 μm, the hole width in *Y*-axis direction *w_Ry_* = 5.5 μm, and the minimum wall width *ρ_R_* = 2.5 μm. After simulation, their IRAs and geometric parameters of the best cases for the six ESCSs are obtained and listed in Table 7. The best case among these individual bests is still the type of three ECs. For four types of ESCSs, the variances of the IRAs with different $D_{x}$ and $D_{y}$ or $W_{x}$ and $W_{y}$ are still similar. Therefore, the best type is still taken as an example to show the variances of the IRAs with different $D_{x}$ and $D_{y}$, and is shown in Figure 19. It is confirmed again that the subwavelength rectangular-hole arrays with rectangular-columnar or elliptical-columnar structures in the hole array can be enhanced the absorption efficiency of this thermopile. Therefore, we try to enhance the best geometry case in Section 4.1 by adding such structures in hole arrays. The results are shown in Table 8.

## 6. Sub-Wavelength Elliptical Hole Array (SEHA)

Comprehensively comparing the results of the above-mentioned best geometric shapes of different hole shapes under different conditions, we find that, under similar conditions, round holes are better than square holes. In addition, the best rectangular holes are better than square holes. Therefore, we speculate that the optimal geometry of the subwavelength elliptical hole array will be better than SRHA. The shape of elliptical hole is similarly to the center section of Figure 18c. Its geometric parameters are represented by symbols similar to circular hole, that is, its hole widths in the *X*-axis and *Y*-axis directions are respectively represented by *d_E__x_* and *d_E__y_*, respectively, and the minimum width of the wall are represented by *ρ_E_*. Here *ρ_E_* is taken as the same as *ρ_C_* for the case where there is no minimum structure line width limit, that is, *ρ_E_* = 1.25 μm in SA type and *ρ_E_* = 1.05 μm in HA type. Figure 20 show the variances of the IRA with different *d_Ey_* by the function of *d_Ex_* for the thermopiles with SA type (a) and HA type (b) of SEHA. It is seen from Figure 20 that, for SA type, the best IRA is about 96.46% at *ρ_R_* = 1.25 μm, *w_Rx_* = 15.8 μm, and *w_Ry_* = 4.2 μm, and, for HA type, the best IRA is about 97.68% at *ρ_R_* = 1.05 μm, *w_Rx_* = 15.9 μm, and *w_Ry_* = 5.3 μm. Comparing the SEHA results with the SRHA results, the SEHA case are indeed better than the SRHA case. Next, besed on minimum structure line width limit of 2.5 μm, the variances of the IRA with different dEy by the function of dEx for the thermopiles with SA type (a) and HA type (b) of SEHA are shown in Figure 21. It is obtained that the IRAs of SA type and HA type are about 90.02% at *w_Rx_* = 15.9 μm, and *w_Ry_* = 4.6 μm and 95.43% at *w_Rx_* = 16.0 μm, and *w_Ry_* = 5.1 μm, and the results of SEHA case is also better than those of SRHA case.

## 7. Discussion of Simulation Results

In order to facilitate the comparisons of the better cases for the above research results, their IRA, relative IAEs, and geometric parameters are summarized in Table 9. Here the relative IAE defined as the IAE of the thermopile with the SHA at the target temperature of 60 °C is relative to one without any SHA in its active area to appear the influence of the CMOS compatible thermopiles with those SHAs and is written as
(5)$$Relative\;IAE\;=\frac{\mathrm{IAE}\;\mathrm{of}\;\mathrm{that}\;\mathrm{thermopile}\;\mathrm{with}\;\mathrm{the}\;\mathrm{SHA}}{\mathrm{IAE}\;\mathrm{of}\;\mathrm{that}\;\mathrm{thermopile}\;\mathrm{without}\;\mathrm{any}\;\mathrm{SHA}}$$

It is seen from Table 9 that the best case among all cases studied is the HA-SRHA with three ECs in all rectangular holes and the relative IAE is about 3.629, that is, the IAE is up to 3.629 times higher than that without any SHA. From the IRAs of those better cases in Table 9 are greater than 96.5%, it can be seen that they are all close to the ideal 100%, and their Fresnel reflections on the interface between SiO_2_ and air are obvious suppressed.

According to the above research results, four interesting structural features or trends are obtained. The first is the hexagonal arrangement results are better than the square arrangement results. There are obvious differences between them, but in the cases of the best structure without any minimum structure line width restriction, the differences between them is not obvious, only some minor differences remain. The second is that the case of rectangular hole is better than the case of square hole, the case of circular hole is better than the case of square hole, and the case of elliptical hole is better than the case of rectangular hole. The relative difference between them gradually becomes smaller with the optimization of their respective structures and the relaxation of the minimum structure line width restrictions. In other words, the IRAs of specific asymmetric rectangle and elliptical hole structure arrays are higher than those of relatively symmetric square and circular hole structure arrays. And when there is no minimum structure line width limit, their optimal values are close to the same. The third is that in the relatively better rectangular hole array, keeping proper materials in those holes are still helpful to the infrared absorbance, and the geometry of the retained material still has a significant effect on the infrared absorbance. Although we currently have no theory to verify our simulation results, the results of trial production experiments that have been conducted are consistent with the trend of simulation results. The fourth is the Fresnel reflection on the interface between SiO_2_ and air is suppressed when those SHAs exist. It can be seen that the IRAs of those better cases are close to 100%.

The current trial-manufacturing process we are using has the limitation of the minimum structural line width, which makes us unable to effectively provide trial-manufacturing verification of those excellent simulation suggestions with smaller structural line widths. In the future, we will study the trial production verification process, which can effectively produce those excellent simulation suggestions with smaller structural line widths. And those better designed SEHAs at minimum wall width of 2.5 μm will be also prepared to fabricate test samples.

## 8. Conclusions

In this study, the thermopiles with various SHA are numerically and experimentally investigated to understand the influence of those SHAs in active area of the thermopile device. Based on the FDTD method, the simulation tool is set up and verified. The prototypes are manufactured by the 0.35 μm 2P4M CMOS-MEMS process in TSMC. The measurement results of those prototypes are similar to their simulation results. Based on the simulation technology, more sub-wavelength hole structural effects for IRA of such thermopile device are discussed. It is found that the HA type results are significantly better than the SA type results and the IAEs of specific asymmetric rectangle and elliptical hole structure arrays are higher than the relatively symmetric square and circular hole structure arrays and the overall best results are respectively up to 3.532 and 3.573 times higher than that without sub-wavelength structure at the target temperature of 60 °C when the minimum structure line width limit of the process is ignored. The experimental results of infrared absorbance for the thermopiles with SHA are greatly increased than one without any SHA. Obvious, the IRA can be enhanced when the SHAs are considered in active area of the thermopile device and the structural optimization of the SHAs is absolutely necessary. Although, the current trial-manufacturing process has the limitation of the minimum structural line width, which makes us unable to effectively provide trial-manufacturing verification of those excellent simulation suggestions with smaller structural line widths. In the future, if there are processes that can effectively manufacture those excellent simulation suggestions with smaller structural line widths, they can provide reference for trial production.

## Figures and Tables

**Figure 1 sensors-21-00180-f001:**
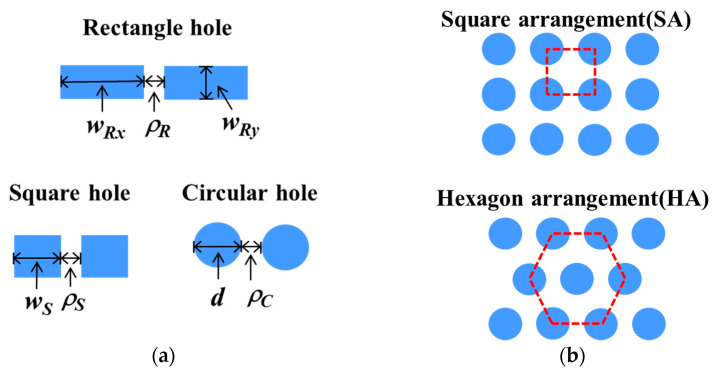
Top-view sketches of (**a**) three investigated hole shapes and (**b**) the hole arrangements described by use of circular holes as an example.

**Figure 2 sensors-21-00180-f002:**
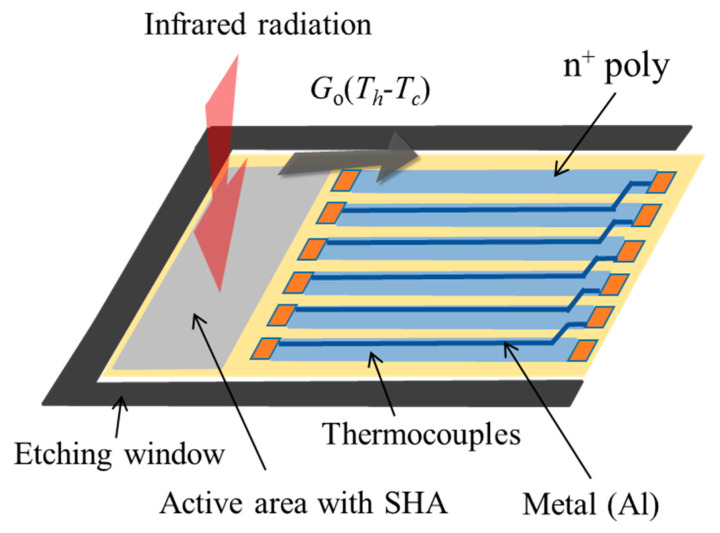
Sketch of the designed CMOS compatible thermopile configuration.

**Figure 3 sensors-21-00180-f003:**
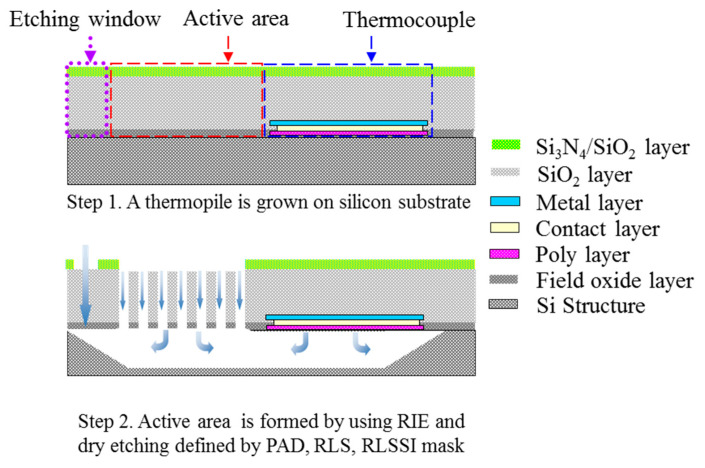
Schematic diagram of the structure of TSMC 0.35 µm 2P4M CMOS-MEMS process.

**Figure 4 sensors-21-00180-f004:**
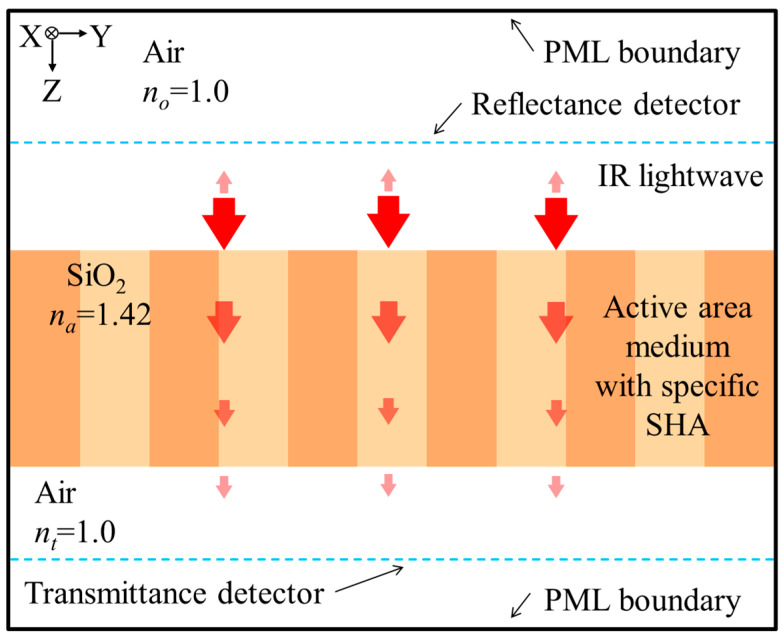
Sketch of a light-wave propagation through a CMOS compatible thermopile with specific SHA simulated by the FDTD method.

**Figure 5 sensors-21-00180-f005:**
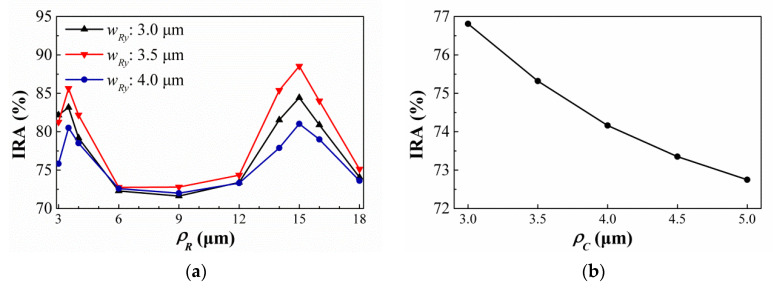
Variances of the IRA by the function of minimum wall width for the thermopiles with (**a**) SRHA and (**b**) SCHA, and the thermopile without any SHA.

**Figure 6 sensors-21-00180-f006:**
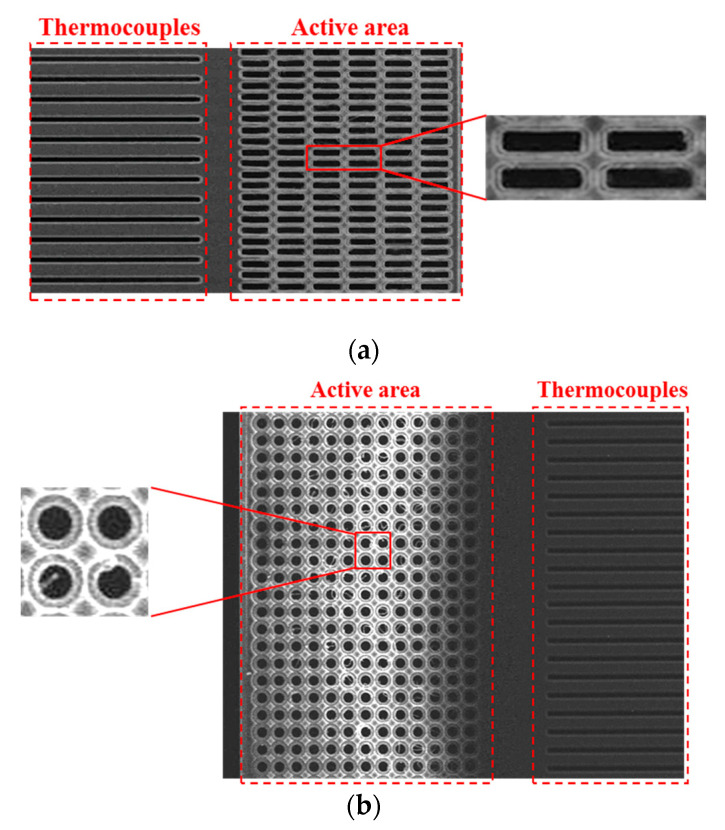
SEM image of the CMOS compatible thermopile with (**a**) SRHA of 3.5 μm case and (**b**) SCHA of 3 μm case.

**Figure 7 sensors-21-00180-f007:**
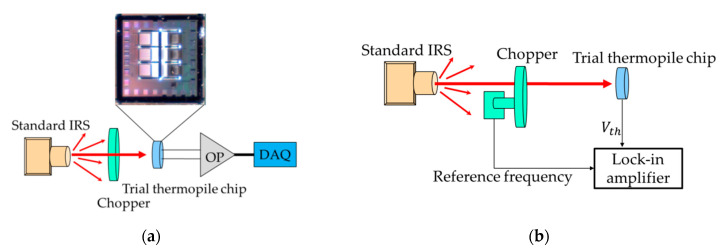
Setups of the experimental measurement framework of the output voltage (**a**) and the frequency response (**b**) for the trial thermopiles.

**Figure 8 sensors-21-00180-f008:**
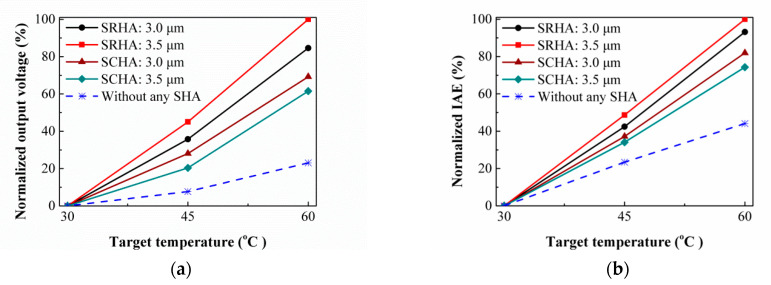
(**a**) Normalized output voltages and (**b**) normalized IAEs with the target temperature for the thermopiles with SRHA and SCHA, and without any SHA.

**Figure 9 sensors-21-00180-f009:**
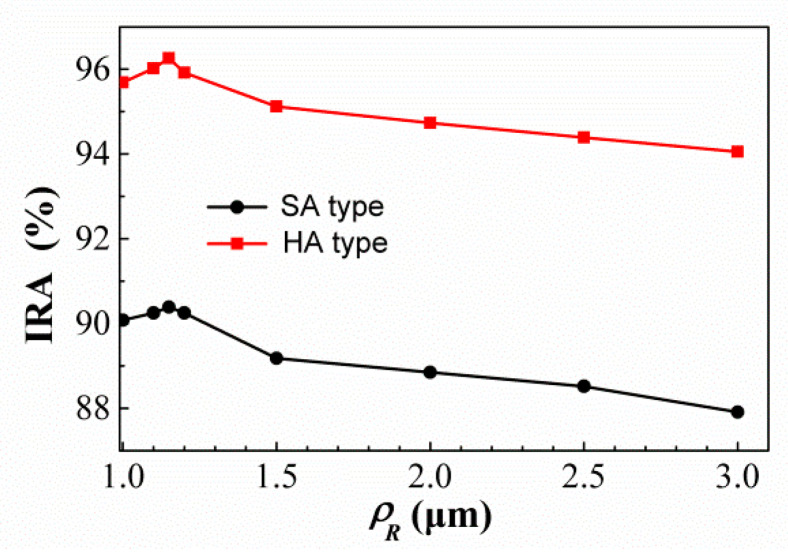
Variances of the IRA for the thermopiles with SA and HA types of SRHA by the function of *ρ_R_* at the hole widths of 15.5 μm and 5.5 μm.

**Figure 10 sensors-21-00180-f010:**
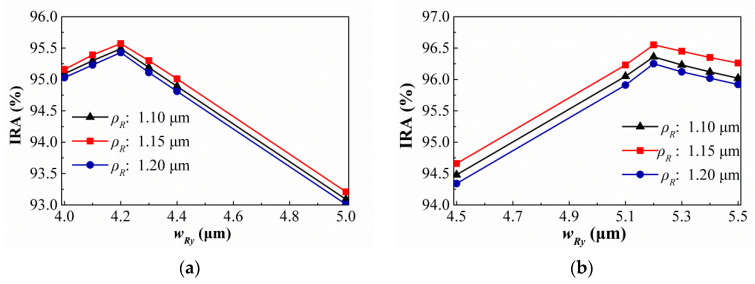
Variances of the IRA with different *ρ_R_* by the function of *w_Ry_* for the thermopiles with SA type (**a**) and HA type (**b**) of SRHA of *w_Rx_* = 15.5 μm.

**Figure 11 sensors-21-00180-f011:**
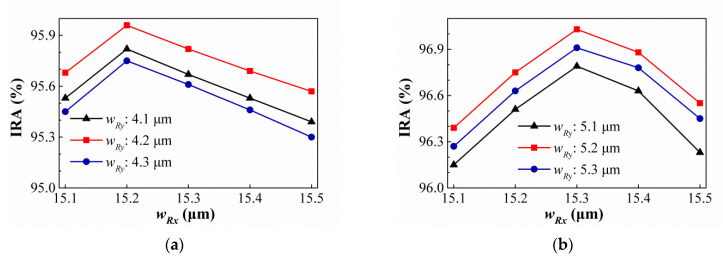
Variances of the IRA with different *w_Ry_* by the function of *w_Rx_* for the thermopiles with SA type (**a**) and HA type (**b**) of SRHA of *ρ_R_* = 1.15 μm.

**Figure 12 sensors-21-00180-f012:**
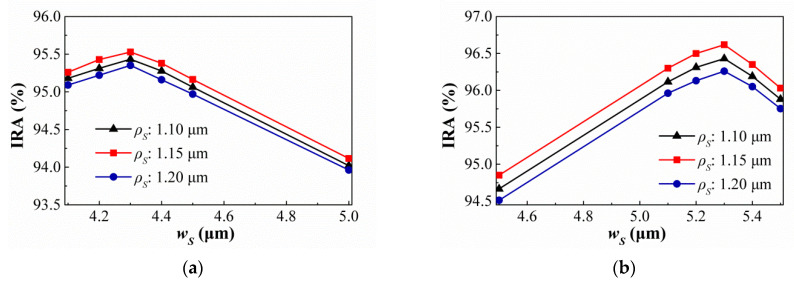
Variances of the IRA with different *ρ_S_* by the function of *w_S_* for the thermopiles with SA type (**a**) and HA type (**b**) of SSHA.

**Figure 13 sensors-21-00180-f013:**
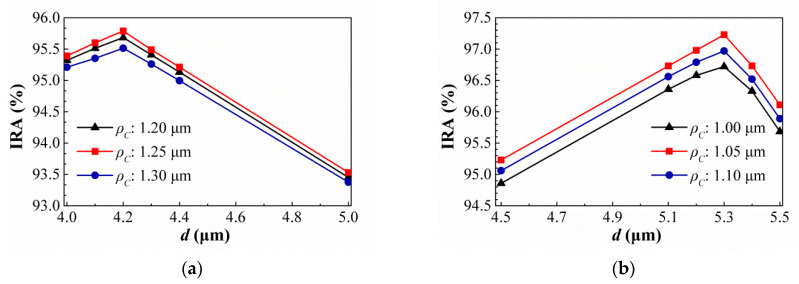
Variances of the IRA with different *ρ_C_* by the function of *d* for the thermopiles with SA type (**a**) and HA type (**b**) of SCHA.

**Figure 14 sensors-21-00180-f014:**
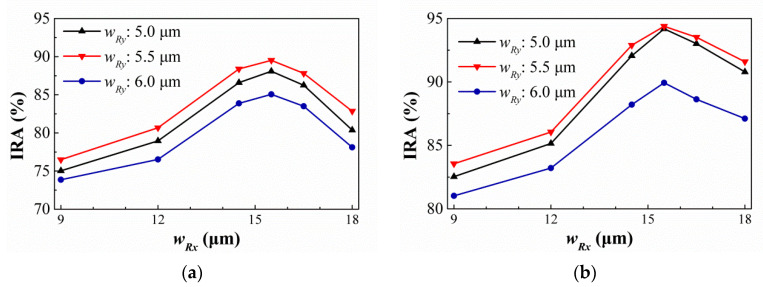
Variances of the IRA with different *w_Ry_* by the function of *w_Rx_* for the thermopiles with SA type (**a**) and HA type (**b**) of SRHA.

**Figure 15 sensors-21-00180-f015:**
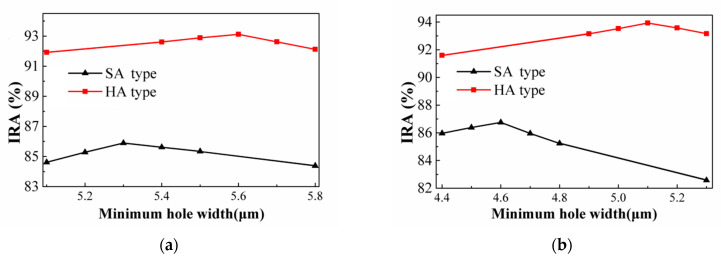
Variances of the IRA by the function of minimum hole width, *w_S_* or *d*, for the thermopiles with SA and HA types of (**a**) SSHA and (**b**) SCHA at minimum wall width of 2.5 μm.

**Figure 16 sensors-21-00180-f016:**
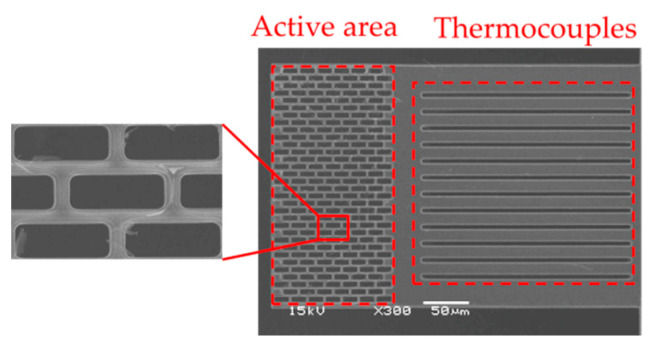
HA-SRHA SEM image of the CMOS compatible thermopile with HA-SRHA of *w_Rx_* = 15.5 μm and *w_Ry_* = 5.5 μm.

**Figure 17 sensors-21-00180-f017:**
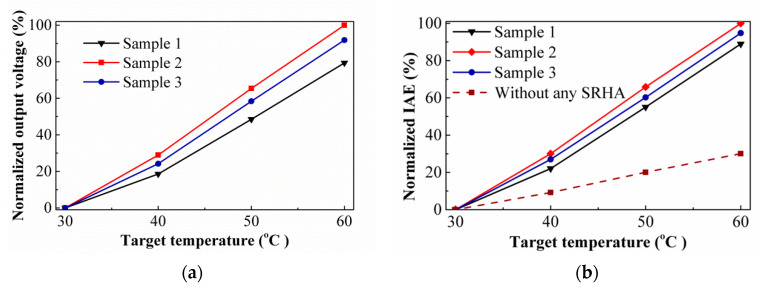
Normalized output voltages (**a**) and normalized IAEs (**b**) with the target temperature for three trial thermopiles.

**Figure 18 sensors-21-00180-f018:**
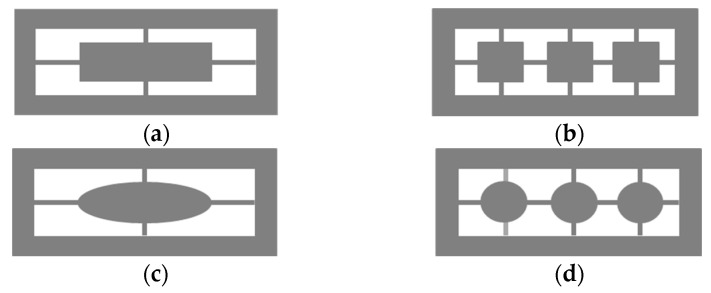
Top-view sketch of the six ESCSs, (**a**) one RC, (**b**) three RCs, (**c**) one EC, and (**d**) three ECs.

**Figure 19 sensors-21-00180-f019:**
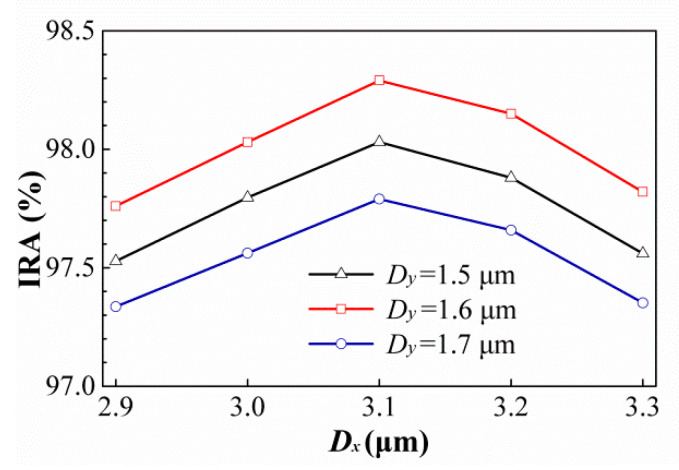
Variances of the IRAs with different $D_{x}$ and $D_{y}$ for the thermopiles with three ECs type in HA rectangular holes *ρ_R_ =* 2.5 μm.

**Figure 20 sensors-21-00180-f020:**
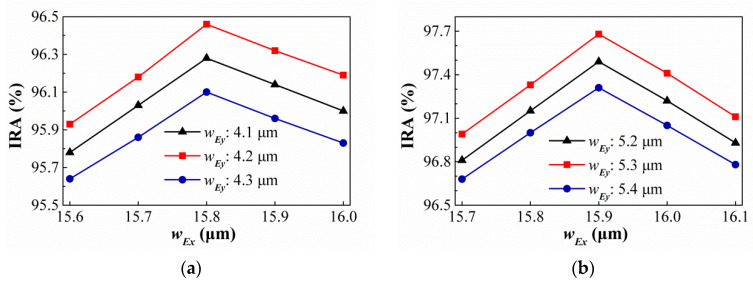
Variances of the IRA with different *d_E__y_* by the function of *d_E__x_* for the thermopiles with SA type (**a**) and HA type (**b**) of SEHA when there is no minimum structure line width limit.

**Figure 21 sensors-21-00180-f021:**
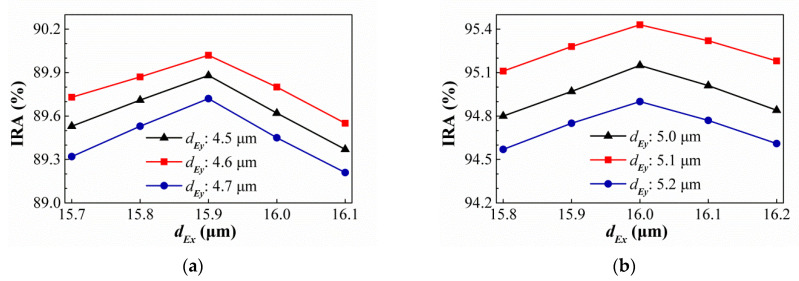
Variances of the IRA with different *d_E__y_* by the function of *d_E__x_* for the thermopiles with SA type (**a**) and HA type (**b**) of SEHA when the minimum structure line width limit is 2.5 μm.

**Table 1 sensors-21-00180-t001:** Range and interval of geometric parameters selected during simulation.

Hole Shape	Minimum Wall Widths (μm)		Hole Widths (μm)			
			X-Direction		Y-Direction	
	Range (μm)	Interval (μm)	Range (μm)	Interval (μm)	Range (μm)	Interval (μm)
Rectangular	3–5	0.5	3–18	1(3) *	3–5	0.5
Circular	3–5	0.5	3–5	0.5	-	-

* The interval of preliminary search is 3 μm and the interval of detailed search 1 μm.

**Table 2 sensors-21-00180-t002:** Geometric parameters of five trial-produced samples.

Sample Number	Hole Shape	Minimum Wall Widths (μm)	Hole Widths (μm)	
			X-Direction	Y-Direction
1	Rectangular	3.0	15.0	3.0
2		3.5	15.0	3.5
3	Circular	3.0	3.0	3.0
4		3.5	3.5	3.5
5	Without any SHA			

**Table 3 sensors-21-00180-t003:** Measurement results of the SHA geometric parameters and modified simulation results for five trial samples.

Sample Number	Average Wall Widths (μm)		Average Hole Widths (μm)		Normalized IAEs
	X-Direction	Y-Direction	X-Direction	Y-Direction	
1	2.85	2.89	15.15	3.27	93.20
2	3.26	3.29	15.21	3.83	100.00
3	2.93	2.89	3.16	3.21	81.96
4	3.32	3.28	3.82	3.90	74.31
5	Without any SHA				44.11

**Table 4 sensors-21-00180-t004:** Geometric parameters, simulated IRAs, and normalized simulated IAEs at target temperature 60 °C for the thermopiles with various optimal SHA.

Pattern Type		Geometric Parameters (μm)			IRA (%)	Normalized IAE (%)
Arrangement	Hole Type	Minimum Wall Widths	Hole Widths			
			X-Direction	Y-Direction		
SA	SRHA	1.15	15.2	4.2	95.96	96.60
	SSHA	1.15	4.3	4.3	95.53	95.45
	SCHA	1.25	4.2	4.2	95.97	96.63
HA	SRHA	1.15	15.3	5.2	97.23	100.00
	SSHA	1.15	5.3	5.3	96.62	98.37
	SCHA	1.05	5.3	5.3	97.03	99.46

**Table 5 sensors-21-00180-t005:** Geometric parameters, simulated IRAs, and normalized simulated IAEs for the thermopiles with various optimal SHA at minimum wall width of 2.5 μm.

Pattern Type		Hole Widths (μm)		IRA (%)	Normalized IAE (%)
Arrangement	Hole Type	X-Direction	Y-Direction		
SA	SRHA	15.5	5.5	89.52	86.57
	SSHA	5.3	5.3	85.89	76.56
	SCHA	4.6	4.6	86.75	78.93
HA	SRHA	15.5	5.5	94.39	100.00
	SSHA	5.6	5.6	93.12	96.50
	SCHA	5.1	5.1	93.93	98.73

**Table 6 sensors-21-00180-t006:** Measurement results of the HA-SRHA geometric parameters and modified simulation results for three proposed thermopiles.

Sample Number	Average Wall Widths (μm)		Average Hole Widths (μm)		Normalized IAE (%)
	*ρ_Rx_*	*ρ_Ry_*	*w_Rx_*	*w_Ry_*	
1	2.40	2.53	12.60	5.47	86.73
2	2.40	2.49	15.60	5.51	100
3	2.39	2.51	18.61	5.49	92.55

**Table 7 sensors-21-00180-t007:** Their IRAs and geometric parameters of the best cases for the four ESCSs in rectangular holes *ρ_R_ =* 2.5 μm of the [52] best case.

ESCS Type	Geometric Parameters		IRA (%)
	$W_{x}/D_{x}\;(\mu m)$	$W_{y}/D_{y}\;(\mu m)$	
Without Any ESCS	-	-	94.39
One RC	11.5	1.5	97.59
Three RCs	3.3	1.9	97.96
One EC	12.1	1.5	97.86
Three ECs	3.1	1.6	98.29
Geometric parameters of the HA-SRHA: *ρ_R_* = 2.5 μm, *w_Rx_* = 15.5 μm and *w_Ry_* = 5.5 μm			

**Table 8 sensors-21-00180-t008:** Their IRAs and geometric parameters of the best cases for the four ESCSs in rectangular holes *ρ_R_ =* 1.15 μm of the best case of Section 4.1.

ESCS Type	Geometric Parameters		IRA (%)
	$W_{x}/D_{x}\;(\mu m)$	$W_{y}/D_{y}\;(\mu m)$	
Without Any ESCS	-	-	97.27
One RC	11.8	1.5	97.79
Three RCs	3.2	1.5	98.26
One EC	12.2	1.0	98.02
Three ECs	3.3	1.3	98.61
Geometric parameters of the HA-SRHA: *ρ_R_* = 1.15 μm, *w_Rx_* = 15.3 μm and *w_Ry_* = 5.2 μm			

**Table 9 sensors-21-00180-t009:** Their IRA, relative IAEs, and geometric parameters of the better cases in this study.

Case	Hole Type	Minimum Wall Widths (μm)	Hole Widths (μm)		With or Without ESCS	IRA (%)	Relative IAE
			X-Direction	Y-Direction			
1	HA-SSHA	1.15	5.3	5.3	Without	96.62	3.477
2	HA-SCHA	1.05	5.3	5.3	Without	97.03	3.514
3	HA-SEHA	1.05	15.9	5.3	Without	97.68	3.573
4	HA-SRHA	1.15	15.3	5.2	Without	97.27	3.532
5					With three RCs	98.26	3.626
6					With three ECs	98.61	3.658
7	HA-SRHA	2.5	15.5	5.5	Without	94.39	3.252
8					With three RCs	97.96	3.599
9					With three ECs	98.29	3.629

## Data Availability

The data presented in this study are available on request from the corresponding author.

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
