# Peer review of "A Thermopile Device with Sub-Wavelength Hole Arrays by CMOS-MEMS Technology"

_sensors, 2020, doi:10.3390/s21010180_

Round 1

Reviewer 1 Report

Abstract: “the infrared absorption efficiency (IAE) of hexagonal hole is higher than the other researched 22 holes and the overall best result is up to 3.532 times higher than that without sub-wavelength 23 structure”. How much of the improvement in response is due to better thermal isolation and how much to optical effects e.g. increased absorption?

Can the authors provide more details about the experimental setup used for the responsivity measurements e.g. type of target used, wavelength range covered?

Can the authors comment on the spectral dependence of the device?

Simulations are discussed but no significant results are presented. Can the authors’ add the simulated results for comparison?

Line 295: I do not believe the AD8551 amplifier is chopper stabilized, although it has very low drift.

Line 130: “The investigated SHA patterns 130 are three types of hole shapes including rectangular, square, circular, and hexagonal, referred to as SRHA, SSHA, and SCHA” I make this 4 types. Can the authors’ clarify?

Reviewer 2 Report

The content of the article and the presented measurement data show that in the experimental tests one measurement was performed for a given configuration.

One cannot draw too far-reaching conclusions if the entire series of measurements has not been carried out for one configuration and with the same input parameters.

Reviewer 3 Report

This manuscript presents simulation and experimental development of a thermopile with various shaped sub-wavelength hole arrays (SHA) and finds that hexagonal-shaped SHA yields the best performance.

The manuscript is well written and the results are conclusive.

My only concern here is- it seems there is a significant work overlap with the already published work here- https://www.mdpi.com/2076-3417/9/23/5118

Could authors please justify this?

Reviewer 4 Report

Manuscript 1016295 is a more systematical description of a couple of pieces of work the authors have published recently in other journals. Upon review of some related literature, the reviewer feel that the following aspects need to be improved and addressed before the manuscript can be accepted for publication.  

Context

The additional information in this manuscript, which can be considered as an extension from what the authors have published, is incremental. Fundamental theoretical and designing considerations have been discussed in the previous work, especially in reference [52] and [53]. Therefore, it’s suggested that the manuscript, at least the Introduction section, can be reorganized to reflect the previous discussions. Considering a special technology – TSMC 0.35um – is used for the device implementation, due to the material and dimensional limits, the introduction of thermopile structures can be more quickly converged to the demonstrated structure that employs polysilicon and aluminum stripes.

There are too many combinations of geometric parameters – sizes, shapes of the transmission holes array. Some special designs are presented on top of these combinations, simply to make the manuscript very lengthy. It’s suggested that the authors simplify the discussions by presenting only major parameters, or, referring some discussions to the previously published two papers.

Technical Discussions

While the authors have broadly discussed the optimal in-plane geometric designs, they left out some useful information related to the spatial arrangement of the device and measurements. As a CMOS technology (TSMC35 2P4M) and dry-etching based post-CMOS microfabrication steps are involved in the device implementation, discussions on the effects of some process variations would be beneficial to the readers’ reference. The reviewer is interested in the following technical details.

  • How much warp or curling of the active and thermopile areas was observed after the device fabrication? A surface profilometry may be needed to examine the surface morphology that may affect the performance of the SHAs.
  • If there is any obvious deformation, how will it affect the efficiency of the absorbance? If the deformation is significant, will Fabry-Perot like effect be developed?
  • Is it possible to simulate the IRA variations caused by such deformation in FDTD simulator?
  • It’s necessary to explain “RLS” and “RLSSI” process as they may be confused with other acronyms.

Writing

The overall writing of the manuscript is acceptable. However, there are still many grammatical errors and typos to be corrected. For instance,

  • Line 128, the sentence is grammatically inappropriate.
  • In Line 236, there are typos related “simulation”.
  • The reference correspondence should be correct. For example, Reference 31 doesn’t seem to be correctly cited.
  • The contrast of Fig. 6(a) and (b) can be increased. The image is almost totally blackout in printouts. Meanwhile, the color tone in Fig. 18 can be changed to have stronger contrast. The printout of the subfigures in Fig. 8 shows some washout.
  • Even the first name of the first author doesn’t seem correct.

The reviewer would be glad to provide a second round review of the manuscript should the authors concur with the reviewer on the above suggestions.          

Round 2

Reviewer 3 Report

I am happy with authors's response, thanks. 

Reviewer 4 Report

Thank you for addressing many of the concerns this reviewer has raised for the previous version of the manuscript. I understand your lack of instruments for additional tests and analysis proposed by the reviewer. Hopefully you can continue your work for future improvements.

Meanwhile, it’s suggested that a better publication plan be developed so that the future papers can have appropriate citations to avoid tedious descriptions, especially when incremental findings are presented.

With these being said, the reviewer agrees that this manuscript can be accepted for publication.
